# Neural Network-Based Adaptive Height Tracking Control of Active Air Suspension System with Magnetorheological Fluid Damper Subject to Uncertain Mass and Input Delay

**DOI:** 10.3390/s24010156

**Published:** 2023-12-27

**Authors:** Rongchen Zhao, Haifeng Xie, Xinle Gong, Xiaoqiang Sun, Chen Cao

**Affiliations:** 1School of Mechanical and Electrical Engineering, Guizhou Normal University, Guiyang 550001, China; zhaorongchen@outlook.com (R.Z.); taylor_hai@163.com (H.X.); 2School of Vehicle and Mobility, Tsinghua University, Beijing 10084, China; xinlegong@gmail.com; 3Automotive Engineering Research Institute, Jiangsu University, Zhenjiang 212013, China; sunxqujs@126.com; 4Research Institute of Highway, Ministry of Transport, Beijing 100088, China

**Keywords:** MRD-AAS system, ride height control, RBFNN approximator, uncertain sprung mass, time-varying input delay

## Abstract

In this paper, we present a novel robust adaptive neural network-based control framework to address the ride height tracking control problem of active air suspension systems with magnetorheological fluid damper (MRD-AAS) subject to uncertain mass and time-varying input delay. First, a radial basis function neural network (RBFNN) approximator is designed to compensate for unmodeled dynamics of the MRD. Then, a projector-based estimator is developed to estimate uncertain parameter variation (sprung mass). Additionally, to deal with the effect of input delay, a time-delay compensator is integrated in the adaptive control law to enhance the transient response of MRD-AAS system. By introducing a Lyapunov–Krasovskii (LK) functional, both ride height tracking and estimator errors can robustly converge towards the neighborhood of the desired values, achieving uniform ultimate boundness. Finally, comparative simulation results based on a dynamic co-simulator built in AMESim 2021.2 and Matlab/Simulink 2019(b) are given to illustrate the validity of the proposed control framework, showing its effectiveness to operate ride height regulation with MRD-AAS systems accurately and reliably under random road excitations.

## 1. Introduction

The growing need to isolate vibrations caused by uneven road surfaces has led to more concentration on the vehicle suspension system, which is considered a vital component of the chassis in automotive manufacturing companies, such as Li Auto and Tesla. The electronically controlled active air suspension (AAS) systems adjust the ride height and posture movements by inflating and deflating the air spring so as to protect the vehicle body on bumpy roads and reduce air drag. In connection with the magnetorheological fluid damper, the AAS systems are more effective in enhancing ride comfort and vehicle maneuverability during complex driving conditions as compared with conventional passive and semi-active suspensions. The active air suspension systems with magnetorheological fluid damper (MRD-AAS) provide an alternative solution to meet the stringent requirements of chassis functions.

Practically, there exist some challenges in maintaining the ride height control under random road excitations robustly and accurately with MRD-AAS systems, since it is difficult to establish a mathematical model to depict the highly nonlinear dynamic characteristics of the air spring and magnetorheological fluid damper (MRD) simultaneously, especially considering the variable stiffness and hysteresis of the adjustable air spring and MRD during the ride height regulation process. Moreover, the uncertain mass caused by the omnipresent changes in payload [1,2] and the effect of input delay induced by pneumatic and hydraulic actuators [3] impose complexities in control system design. These nonlinearities, parametric uncertainties and input delay could severely worsen the ride height tracking control performance or even lead to instability if they are not well addressed.

Presently, a series of models have been established to describe the dynamic characteristics of AAS systems for ride height control. For instance, the authors of [4] analyzed the relationship between spring force and air spring volume and built a mapping model to track the spring force using a back propagation neural network method. However, the modeling accuracy depended on a large amount of experiment tests. The authors of [5] established a nonlinear model of an AAS system on the basis of thermodynamics to represent the dynamic characteristics of the air spring, which are classified as unmodeled dynamics and parametric uncertainties. However, the nonlinear and hysteretic characteristics of the MRD are not fully considered, which would degrade ride height control performance. A number of MRD dynamic models have been constructed to describe the nonlinear and hysteretic characteristics under random road excitations, such as the normalized phenomenological model [6], the hyperbolic tangent model [7], the restructured phenomenological model [8], and the adjustable sigmoid model [9]. However, these models rely largely on the shape functions and a series of parameter identification with optimization techniques. However, by considering the omnipresent parameter variations due to the uncertain sprung mass, it is difficult to maintain the accuracy of the optimized phenomenological model. Therefore, it is important to design a real-time approximator of the phenomenological MRD force to ensure effective ride height control with the MRD-AAS system. Due to the good generalization ability and simplified form, an RBFNN will be considered to design the damper force approximator to make up for the defect of the phenomenological damper model [6,7,8,9].

Despite the challenge caused by the MRD-AAS system modeling for depicting the intrinsic strong nonlinearities and hysteretic characteristics, taking into account the designing control methods for dealing with these characteristics, sensitive parameter variations (such as sprung mass) and actuator input delay are also necessary during the ride height control procedure. In order to handle the highly nonlinear and hysteretic characteristics of the MRD-AAS system, numerous control methods have been proposed, such as sliding mode control, backstepping control, and fuzzy logic and neural network-based control techniques. The authors of [10] proposed a compensated backstepping controller combining traditional backstepping techniques with an adaptive radial basis function. In order to estimate unmodeled dynamic characteristics, fuzzy logic systems and neural networks are employed as the approximators for improving tracking control performance [11,12,13]. However, to our best knowledge, the number of required rules or neurons remains uncertain for making the approximation error stay within a small specific range. In particular, the estimation errors of the uncertain sprung mass were not proofed to converge to zero, theoretically.

Moreover, the MRD-AAS system is composed of the hydraulic and pneumatic actuators, which induce input delay during the ride height controller design procedure. It is also a challenge caused by designing controllers for handling the effect of the input delay, avoiding system instability. Although the delay may be quite short, it can restrict the performance of the controller. To conquer this problem, a Lyapunov–Krasovskii functional is introduced to handle the effect of unknown sufficiently slow time-varying input time lags. The authors of [14] used linear matrix inequalities (LMIs) to solve for the time delay problem of magnetorheological semi-active suspensions and obtained conclusions such as a maximum critical delay of 32.4 (ms) for MRD. The paper [15] investigated the nonlinear dynamics of a constant coefficient time lag vibration control system in detail, and obtained a series of important criteria such as the ability of the system to remain stable in a smaller interval of time delay feedback. The authors of [16] used a locally optimal hybrid PSO/LMI algorithm for perturbations and time delays and tuned the continuous saturation controller by adjusting the feedback gain, but they did not consider the effect of sprung mass variations from the controller. Thus, inspired by the above, a predictor-based compensation scheme is used in this work to address the effects of input delay and uncertainty.

Inspired by the above discussion and considerations, this paper presents a solution to address the height tracking control problem for nonlinear MRD-AAS systems with dynamic characteristics, uncertain parameter variation, and time-varying input delay. The novelties and contributions of this paper are summarized as follows:1.A robust adaptive ride height tracking control framework is proposed, where a MRD force approximator is designed to improve the accuracy of MRD-AAS system modeling, achieving uniform ultimate boundedness in the presence of uncertain sprung mass and time-varying input delay.2.A projector-based nonlinear estimator is developed to estimate the uncertain sprung mass for enhancing the adaptive performance of the proposed control system.3.A time-delay compensator is introduced to deal with the time-varying input delay induced by the pneumatic and hydraulic actuators for improving the robustness of the proposed control system.

With respect to the existing literature, the main merits of the proposed adaptive robust ride height control strategy are as follows. Different from the phenomenological damper model presented in [6,7,8,9], the proposed control system is designed based on a nonlinear mathematical model with the MRD force approximator so as to improve the modeling accuracy of the MRD-AAS system. Compared with the mass estimation method presented in [11,12,13], the developed projector-based mass estimator can ensure that the estimation error is bounded by a certain value, achieving adaptive performance. To improve the anti-interference performance, a time-delay compensator is integrated in the proposed control strategy that allows us to compensate for time-varying input delay. The remainder of this paper is structured as follows.

The organization of this paper is as follows: In Section 2, the notation used throughout this paper is presented. Section 3 involves the dynamic modeling of the MRD-AAS. Section 4 describes the design method for the MRD-AAS system. Section 5 illustrates the effectiveness of proposed control strategy in co-simulation. Finally, Section 6 summarizes the paper.

## 2. Notation

$R^{n}$ denotes the n-dimensional Euclidean space and is used throughout the paper. A function *f* is defined as $R\to R^{n}$ is of class $C^{n}$ if the derivation exists and $f^{{}^{\prime }},f^{{}^{\prime \prime }},\ldots ,f^{n}$ are continuous. For your reference, Table 1 lists the main symbols and descriptions of the model, controller, and estimator parameters as follows.

## 3. Problem Formulation

The objective of this section is to formulate the problem of ride height tracking control with application to the MRD-AAS suspension system. As displayed in Figure 1, there are several components among the sprung mass and unsprung mass of a quarter vehicle MRD-AAS suspension, such as an adjustable air spring, an MRD, and other accessories. We start by describing the modeling of a quarter vehicle with MRD-AAS by considering the variable passenger or/and payload, time lags induced by the pneumatic and hydraulic actuators, and disturbance. Then, the issue of maintaining ride height tracking control is formulated by taking into account the uncertain sprung mass, time-varying input delay, and disturbance.

### 3.1. MRD-AAS System Modeling

For addressing the problem of the ride height tracking control, based on the modified Binghan mathematical model for magnetorheological dampers, a mathematical model of a quarter vehicle with MRD-AAS is established. The schematic diagram of this quarter vehicle with MRD-AAS is displayed in Figure 1, and based on Newton’s second law, the dynamical equations of the masses are expressed by
(1)$$\left\{\begin{array}{c} m_{s}{\ddot{z}}_{s}=F_{as}-F_{d}-F_{g}\\ m_{u}{\ddot{z}}_{u}=-F_{as}+F_{d}-F_{k}-D_{t}, \end{array}\right.$$
where $F_{as}$ denotes the air spring force; $F_{d}$ represents MRD damper force; $F_{g}$ is the gravitational force; $z_{s}$ and $z_{u}$ are the sprung mass and unsprung mass displacement of a quarter vehicle; and $m_{s}$ and $m_{u}$ are the sprung mass and unsprung mass (the wheel assembly). $F_{t}$ and $D_{t}$ are the elasticity force and damping force of the tire, respectively. Forces produced by the adjustable air spring and the tire are given as
(2)$$\left\{\begin{array}{c} F_{as}={\bar{A}}_{as}\left(p_{as}-p_{atm}\right)\\ F_{k}=k_{t}\left(z_{u}-z_{r}\right)\\ D_{t}=c_{t}\left({\dot{z}}_{u}-{\dot{z}}_{r}\right), \end{array}\right.$$
where $p_{atm}$ denotes the atmospheric pressure, $s_{p}=10^{5}$, and ${\bar{A}}_{as}=A_{as}s_{p}$ is the effective area of the adjustable air spring. $z_{r}$ denotes the random road excitation, and $k_{t}$ and $c_{t}$ are the stiffness and damping coefficient of the tire, respectively. According to the performance tests of the magnetorheological hysteresis suspension presented in [17,18], a hyperbolic tangent model is employed to effectively describe the nonlinear hysteretic characteristics of the MRD current force, expressed by
(3)$$F_{d}=ftanh\left(\alpha {\dot{z}}_{s}+\beta sign\left(z_{s}\right)\right)+c{\dot{z}}_{s},$$
where
(4)$$\left\{\begin{array}{c} f=f_{1}I+f_{2}\\ \alpha =\alpha_{1}I+\alpha_{2}\\ \beta =\beta_{1}I+\beta_{2}\\ c=c_{1}I+c_{2}, \end{array}\right.$$
where $f,\;f_{1},\;f_{2}$ denote the hysteresis loop scaling factors; $\alpha ,\;\alpha_{1},\;\alpha_{2}$ are the hysteresis loop slope scaling factors, $\beta ,\;\beta_{1},\;\beta_{2}$ are the half-width factors of the hysteresis loop, *I* is the current, and $c,\;c_{1},\;c_{2}$ are the MRD damping coefficients.

Due to the inherent nonlinearities and uncertainties in the MRD-AAS system, the modeling lumped error, time-varying input delay, and disturbance should be considered in the employed model for ride height control. Following [5], the MRD-AAS is expressed as
(5)$$\left\{\begin{array}{c} {\dot{x}}_{1}=x_{2}\\ {\dot{x}}_{2}=m_{s}^{-1}\left({\bar{A}}_{as}\left(x_{3}-p_{atm}\right)-ftanh\left(\alpha x_{2}+\beta sign\left(x_{1}\right)\right)-cx_{2}-F_{\delta }\right)-g\\ {\dot{x}}_{3}=u_{\tau }-x_{2}x_{3}A_{as}\gamma v_{as}^{-1}+\left(\gamma -1\right){\left(s_{p}v_{as}\right)}^{-1}\dot{Q}+d_{n}\left(t\right), \end{array}\right.$$
where $x_{1}=z_{s}$, $x_{2}={\dot{z}}_{s}$, $x_{3}=p_{as}/s_{p}$, $v_{as}$ is air volume, $\dot{Q}$ is the heat transfer rate, and γ denotes the specific heat ratio. $u_{\tau }$ denotes the delayed control input, $d_{n}\left(t\right)$ is external time-varying disturbance, satisfying $\left|d_{n}\left(t\right)\right|\leq {\bar{d}}_{n}$ with ${\bar{d}}_{n}$ denoting a known positive constant, and $F_{\delta }$ represents the concentrated modeling error of the MRD. Details of the nonlinear active air suspension system modeling can be found in [5].

**Remark 1.** 
*By considering the practical operating scenarios of ride height tracking control with MRD-AAS systems, the vertical displacement of sprung mass $x_{1}$, velocity $x_{2}$, and $x_{3}$ could be measured by the configured signal acquisition devices, e.g., the vehicle-equipped displacement and pressure sensors.*


**Assumption 1.** 
*Under the same vertical excitation, the air spring force $F_{as}$, the MRD force $F_{d}$, and their derivatives are bounded and satisfy*

(6)
$$\begin{array}{c} |F_{as}|\;\leq {\bar{F}}_{as},\;|F_{d}|\;\leq {\bar{F}}_{d},\;|{\dot{F}}_{as}|\;\leq {\bar{F}}_{at},\;\left|{\dot{F}}_{d}\right|\leq {\bar{F}}_{dt}, \end{array}$$

*where ${\bar{F}}_{as},\;{\bar{F}}_{d},\;{\bar{F}}_{at}$, and ${\bar{F}}_{dt}$ are known positive constants [11].*


**Assumption 2.** 
*By considering the omnipresent changes of passenger or/and payload, the sprung mass $m_{s}$ is uncertain, constant and satisfied*

(7)
$$\begin{array}{c} m_{min}\leq m_{s}\leq m_{max}, \end{array}$$

*where $m_{min}$ and $m_{max}$ are known minimum and maximum constants of the vehicle sprung mass.*


**Assumption 3.** 
*The time delay τ is unknown, positive, time-varying, and satisfies $\tau \leq \bar{\tau }$, where $\bar{\tau }$ is a known positive constant. There exists a sufficiently accurate constant $\tau_{k}$, satisfying*

(8)
$$\begin{array}{c} |\tau -\tau_{k}|\leq \tau_{u},\;|u\left(t-\tau \right)-u(t-\tau_{k})|\leq u_{e}, \end{array}$$

*where $\tau_{u}$ and $u_{e}$ are known constants, and $\tau_{k}\leq \tau_{max}$. The MRD-AAS system (Equation 5) stays finite during the period of $\left[t_{0},\;t_{0}+\bar{\tau }\right]$.*


### 3.2. RBFNN Approximator

Since the normalized phenomenological models, such as the hyperbolic tangent model, are not accurate enough to capture the nonlinear and hysteretic characteristics of MRD (Equation 3), a RBFNN approximator is designed to approximate $F_{\delta }$ online, which caused by the unmodeled dynamics of the MRD. Following [19], the hidden Gaussian function $h_{j}$ for the neural net *j* is adopted as
(9)$$h_{j}=\exp(-∥\left(\chi -\phi_{j}\right){∥}^{2}/2b_{j}^{2}),\;j=1,2,...,l,$$
where $\chi ={\left[\chi_{1},\;\chi_{2},\ldots ,\;\chi_{3}\right]}^{T}$ is the input vector, $\phi_{j}=\left[\phi_{1},\;\phi_{2},\ldots ,\;\phi_{n}\right]$ is the center vector of neural net *j* and $b_{j}$ is the width of the Gaussian function that are chosen based on a range of input values [20]. Then, $F_{\delta }$ can be expressed as
(10)$$F_{\delta }\left(t\right)=W^{*T}h+\epsilon ,$$
where $W^{*}$ denotes the desired weights of neural net *j*, $h={\left[h_{1},\;h_{2},\ldots ,\;h_{l}\right]}^{T}$ denotes the hidden layer function, and ϵ is the approximation error. Therefore, approximate smooth function ${\hat{F}}_{\delta }$ can be estimated by the RBFNN as follows
(11)$${\hat{F}}_{\delta }\left(t\right)={\hat{W}}^{T}h,$$
where ${\hat{W}}^{T}$ the weights of the RBFNN [21]. To evaluate the performance of the RBFNN approximator, a performance index function is adopted as
(12)$$E\left(t\right)={\left(F_{d}\left(t\right)+{\hat{F}}_{\delta }\left(t\right)-F_{m}\left(t\right)\right)}^{2}/2,$$
where $F_{m}\left(t\right)$ is the training data obtained from experimental tests of the MRD under the current conditions of I=0∼3(A). By using the gradient descent learning algorithm for effectively adjusting the weights $\Delta w_{j}\left(t\right)$, centers $\Delta \phi_{j,i}\left(t\right)$, and widths $\Delta b_{j}\left(t\right)$ numerically to minimize the squared error function *E* [20], the following equations are given as
(13)$$\left\{\begin{array}{c} \Delta w_{j}\left(t\right)=-℘\partial E/\partial w_{j}=℘\left(F_{d}\left(t\right)+F_{\delta }\left(t\right)-F_{m}\left(t\right)\right)h_{j}\\ \Delta \phi_{j,i}\left(t\right)=-℘\partial E/\partial \Delta \phi_{j,i}=℘\left(F_{d}\left(t\right)+F_{\delta }\left(t\right)-F_{m}\left(t\right)\right)w_{j}h_{j}\left(\chi_{i}-\phi_{ji}\right)/b_{j}^{2})\\ \Delta b_{j}\left(t\right)=-℘\partial E/\partial b_{j}=℘\left(F_{d}\left(t\right)+F_{\delta }\left(t\right)-F_{m}\left(t\right)\right)w_{j}h_{j}(∥\left(\chi -\phi_{j}\right)∥/b_{j}^{3}). \end{array}\right.$$

The momentum factor *ı* domain is 0≤ı≤1, the updated parameters $w_{j},\;\phi_{ji}$, and $b_{j}$ are obtained as
(14)$$\left\{\begin{array}{c} w_{j}\left(t\right)=w_{j}\left(t-1\right)+\Delta w_{j}+ı\left(w_{j}\left(t-1\right)-w_{j}\left(t-2\right)\right)\\ \phi_{ji}\left(t\right)=\phi_{ji}\left(t-1\right)+\Delta \phi_{ji}+ı\left(\phi_{ji}\left(t-1\right)-\phi_{ji}\left(t-2\right)\right)\\ b_{j}\left(t\right)=b_{j}\left(t-1\right)+\Delta b_{j}+ı\left(b_{j}\left(t-1\right)-b_{j}\left(t-2\right)\right). \end{array}\right.$$

To verify the effectiveness of the constructed MRD model, a MRD test bench is fabricated as shown in Figure 2a, which consists of an upper fixture, DC power supply, hydraulic exciter, and MRD. As Figure 2b shows, the measured external characteristics of the employed MRD at the current from 0 to 3(A) are collected and given. Moreover, Figure 3a–h show the comparison of a hyperbolic tangent model with and without the designed RBFNN approximator. It can be found that the designed RBFNN approximator could compensate for the concentrated modeling errors and keep the approximated error in a bound of 20(N). Therefore, the constructed MRD model combined the hyperbolic tangent model with the designed RBFNN approximator, which was then applied in the following control system design.

### 3.3. Problem Statement

Taking into account nonlinearities, parameter variations, and time-varying input delays, we tackle the ride height tracking control problem of MRD-AAS systems, stating that the desired height $x_{d}$ should be located into a curve of class $C^{3}$ or above, with limited temporal derivatives. An arbitrary small neighborhood of the desired height $x_{d}$ may be stabilized at all times by designing a control input $u_{\tau }$ in the presence of unmodeled dynamics, uncertain sprung mass, time-varying input delays, and disturbance such that the ride height $z_{s}$ can always be stabilized in an arbitrarily small neighborhood of its desired height $x_{d}$.

## 4. Adaptive Controller Design

As mentioned in the Introduction, the uncertain sprung mass, time-varying input delays, and highly nonlinear dynamic characteristics of the MRD-AAS system impose difficulties on the height tracking control system design. Based on the established quarter vehicle model with the MRD-AAS (Equation 5), a delay-free ride height control law $u_{\tau }$ is synthesized for guaranteeing ultimate global uniform ultimate boundedness. Figure 4 shows the control block diagram of the proposed control strategy. The on-board sensors can measure the states of the MRD-AAS system $x_{1},\;x_{2}$, and $x_{3}$, as stated in Remark 1. To compensate for the hyperbolic tangent modeling error of MRD, a RBFNN approximator is developed with an adaptive gradient descent learning algorithm. A projector-based estimator is designed to estimate the uncertain sprung mass $m_{s}$ mostly caused by the payload. We start the controller design by defining the height tracking error, given as
(15)$$e_{1}=x_{1}-x_{d},$$
and an auxiliary error term $e_{n}$ as
(16)$$e_{n}=\int_{0}^{t}e_{1}\left(\theta \right)d\theta .$$

Our first Lyapunov candidate function is defined as
(17)$$V_{1}=\frac{1}{2}e_{1}^{2}+\frac{1}{2}\xi^{2},$$
where $\xi =e_{n}+\alpha e_{1}$, whose time derivative yields
(18)$${\dot{V}}_{1}=-W_{1}\left(\xi ,\eta \right)+\eta \left({\dot{x}}_{1}-{\dot{x}}_{d}+k_{1}\eta +\xi \right),$$
where $\eta =\alpha \xi +e_{1}$, $W_{1}\left(\xi ,\eta \right)=k_{1}\eta^{2}+\alpha \xi^{2}$, and $\alpha ,k_{1}$ are positive constants.

Following the backstepping technique procedure, we define a new error $e_{2}$ as
(19)$$e_{2}={\dot{x}}_{1}-{\dot{x}}_{d}+k_{1}\eta +\xi .$$

Constructing a new Lyapunov function candidate by incorporating $e_{2}$, we define
(20)$$V_{2}=V_{1}+\frac{1}{2}e_{2}^{2}.$$

Computing the time derivative of $V_{2}$, we have
(21)$${\dot{V}}_{2}=-W_{2}\left(\xi ,\eta ,e_{2}\right)+e_{2}\left(k_{2}e_{2}+\eta +\varpi \left(F_{as}-F_{d}-{\hat{F}}_{\delta }\right)-g-{\ddot{x}}_{d}+k_{1}\dot{\eta }+\dot{\xi }\right)-e_{2}\varpi {\tilde{F}}_{\delta },$$
where $W_{2}\left(\xi ,\eta ,e_{2}\right)=k_{1}\eta^{2}+\alpha \xi^{2}+k_{2}e_{2}^{2}$, $\varpi =1/m_{s}$, and $k_{2}$ is a positive control gain, ${\tilde{F}}_{\delta }$ is the estimation error of MRD force, given as
(22)$${\tilde{F}}_{\delta }=F_{\delta }-{\hat{F}}_{\delta }=W^{*T}h+\epsilon -{\hat{W}}^{T}h=-{\tilde{W}}^{T}h+\epsilon ,$$
where ${\tilde{W}}^{T}={\hat{W}}^{T}-W^{*T}$. Notice that $\varpi \in [1/m_{max},1/m_{min}]$ is unknown. Then, we introduce the estimated $\hat{\varpi }$, which is bonded by make the following projection function, given as
(23)$$p(\hat{\varpi })=\left\{\begin{array}{cc} \varpi_{max}+\varrho \left(1-\exp(\frac{\varpi_{max}-\hat{\varpi }}{\varrho })\right) & ,\hat{\varpi }\geq \varpi_{max}\\ \hat{\varpi } & ,else\\ \varpi_{min}+\varrho \left(1-exp(\frac{\hat{\varpi }-\varpi_{min}}{\varrho })\right) & ,\hat{\varpi }\leq \varpi_{min} \end{array}\right.$$
where $\varpi_{min}$, and ϱ is an arbitrarily small positive number. Then, (Equation 21) is rewritten as
(24)$$\begin{array}{cc} {\dot{V}}_{2}= & -W_{2}\left(\xi ,\eta ,e_{2}\right)+e_{2}\left(k_{2}e_{2}+\eta +p\left(\hat{\varpi }\right)\left(F_{as}-F_{d}-{\hat{F}}_{\delta }\right)-{\ddot{x}}_{d}-g+k_{1}\dot{\eta }+\dot{\xi }+z_{a}\right)-e_{2}z_{a}\\ & +e_{2}\left(\varpi -p(\hat{\varpi })\right)\left(F_{as}-F_{d}-{\hat{F}}_{\delta }\right)-e_{2}\varpi \left({\tilde{W}}^{T}h+\epsilon \right), \end{array}$$
and $z_{a}$ is defined as
(25)$$z_{a}=\int_{t-\tau_{k}}^{t}u\left(\theta \right)d\theta .$$

The term $\left(\varpi -p(\hat{\varpi })\right)$ and $\left({\tilde{W}}^{T}h+\epsilon \right)$ in the last two terms of (Equation 24) result from the uncertain sprung mass and the concentrated modeling error of the MRD force. A new Lyapunov candidate function is then defined to obtain the estimation law for $\dot{\hat{\varpi }}$ and $\dot{\hat{W}}$, given as
(26)$$V_{2b}=V_{2}+\frac{1}{\gamma_{w}}{\tilde{W}}^{T}\tilde{W}+\frac{1}{\lambda }\int_{0}^{\tilde{\varpi }}\left(p\left(\varpi +\tau \right)-\varpi \right)d\tau ,$$
where $\tilde{\varpi }=\hat{\varpi }-\varpi$ denotes the estimation error for sprung mass. Computing the time derivative of $V_{2b}$, we obtain
(27)$$\begin{array}{cc} {\dot{V}}_{2b}= & -W_{2}\left(\xi ,\eta ,e_{2}\right)+e_{2}\left(k_{2}e_{2}+\eta +p\left(\hat{\varpi }\right)\left(F_{as}-F_{d}-{\hat{F}}_{\delta }\right)-{\ddot{x}}_{d}-g+k_{1}\dot{\eta }+\dot{\xi }+z_{a}\right)-e_{2}z_{a}\\ & +\left(\varpi -p(\hat{\varpi })\right)\left(e_{2}\left(F_{as}-F_{d}-{\hat{F}}_{\delta }\right)-\frac{1}{\lambda }\dot{\hat{\varpi }}\right)-e_{2}\varpi \epsilon -{\tilde{W}}^{T}\left(e_{2}\varpi h+\frac{1}{\gamma_{w}}\dot{\hat{W}}\right). \end{array}$$

We choose the updating law for $\dot{\hat{\varpi }}$ and $\dot{\hat{W}}$ as
(28)$$\dot{\hat{\varpi }}=\lambda e_{2}\left(F_{as}-F_{d}-{\hat{F}}_{\delta }\right),$$
(29)$$\dot{\hat{W}}=-\gamma_{w}e_{2}m_{min}h.$$

Substituting (Equation 28) and (Equation 29) into (Equation 24), we obtain
(30)$$\begin{array}{cc} {\dot{V}}_{2b}= & -W_{2}\left(\xi ,\eta ,e_{2}\right)+e_{2}\left(k_{2}e_{2}+\eta +p\left(\hat{\varpi }\right)\left(F_{as}-F_{d}-{\hat{F}}_{\delta }\right)-{\ddot{x}}_{d}-g+k_{1}\dot{\eta }+\dot{\xi }+z_{a}\right)\\ & -e_{2}z_{a}-e_{2}\varpi \epsilon -{\tilde{W}}^{T}he_{2}\left(m_{min}-\varpi \right)\\ \leq & -W_{2}\left(\xi ,\eta ,e_{2}\right)+e_{2}\left(k_{2}e_{2}+\eta +p\left(\hat{\varpi }\right)\left(F_{as}-F_{d}-{\hat{F}}_{\delta }\right)-{\ddot{x}}_{d}-g+k_{1}\dot{\eta }+\dot{\xi }+z_{a}\right)\\ & -e_{2}z_{a}-e_{2}\epsilon -∥{\tilde{W}}^{T}he_{2}\left(m_{min}-\varpi \right)∥\\ \leq & -W_{2}\left(\xi ,\eta ,e_{2}\right)+e_{2}\left(k_{2}e_{2}+\eta +p\left(\hat{\varpi }\right)\left(F_{as}-F_{d}-{\hat{F}}_{\delta }\right)-{\ddot{x}}_{d}-g+k_{1}\dot{\eta }+\dot{\xi }+z_{a}\right)\\ & -e_{2}z_{a}-e_{2}\epsilon . \end{array}$$

**Remark 2.** 
*The introduced functions $\frac{1}{\gamma_{w}}{\tilde{W}}^{T}\tilde{W}$ and $\frac{1}{\lambda }\int_{0}^{\tilde{\varpi }}\left(p\left(\varpi +\tau \right)-w\right)d\tau$ in (Equation 26) are positive.*


Continuing with the backstepping procedure, we define the next error term as
(31)$$\begin{array}{cc} e_{3}= & k_{2}e_{2}+\eta +p\left(\hat{\varpi }\right)\left(F_{as}-F_{d}-{\hat{F}}_{\delta }\right)-{\ddot{x}}_{d}-g+\dot{\xi }+k_{1}\dot{\eta }+z_{a}. \end{array}$$

Defining the next Lyapunov candidate function by augmenting $e_{3}$ as
(32)$$V_{3}=V_{2b}+\frac{1}{2}e_{3}^{2}.$$
and computing the time derivative of $V_{3}$, we have
(33)$$\begin{array}{cc} {\dot{V}}_{3}= & -W_{3}\left(\xi ,\eta ,e_{2},e_{3}\right)-e_{2}z_{a}+e_{3}{\dot{z}}_{a}-e_{2}\epsilon +e_{3}[k_{3}e_{3}+e_{2}+k_{2}{\hat{\dot{e}}}_{2}+\dot{\eta }+p\left(\hat{\varpi }\right)[{\bar{A}}_{as}(u_{\tau }\\ & -\gamma A_{as}x_{2}x_{3}v_{as}^{-1}+\dot{Q}\left(\gamma -1\right){\left(s_{p}v_{as}\right)}^{-1}+d_{n})-\alpha f{\hat{\dot{x}}}_{2}\left(1-tanh^{2}\left(\alpha x_{2}+\beta sign\left(x_{1}\right)\right)\right)\\ & -c{\hat{\dot{x}}}_{2}-{\dot{\hat{F}}}_{\delta }]+\ddot{\xi }+\dot{p}\left(\hat{\varpi }\right)\left(F_{as}-F_{d}-{\hat{F}}_{\delta }\right)-{\dddot{x}}_{d}+k_{1}\ddot{\eta }], \end{array}$$
where ${\dot{z}}_{a}=u\left(t\right)-u\left(t-\tau_{k}\right)$, $W_{3}\left(\xi ,\eta ,e_{2},e_{3}\right)=k_{1}\eta^{2}+\alpha \xi^{2}+k_{2}e_{2}^{2}+k_{3}e_{3}^{2}$, where $k_{3}$ is a positive control gain, and
(34)$$\left\{\begin{array}{c} {\hat{\dot{x}}}_{2}=p\left(\hat{\varpi }\right)\left(F_{as}-F_{d}-{\hat{F}}_{\delta }\right)-g\\ {\hat{\ddot{e}}}_{1}={\hat{\dot{x}}}_{2}-{\ddot{x}}_{d}\\ {\hat{\dot{e}}}_{2}={\hat{\ddot{e}}}_{1}+k_{1}\dot{\eta }+e_{1}+\alpha {\dot{e}}_{1}. \end{array}\right.$$

We note that $d_{n}$ is an unknown perturbation and cannot be included in the control law, so it is replaced by the following inequality [22], using
(35)$$\left|\Upsilon \right|-\Upsilon tanh\left(\frac{\Upsilon }{\epsilon_{r}}\right)\leq 0.2875\epsilon_{r},$$
where $\epsilon_{r}$ is a positive number. Rewriting (Equation 33), we have
(36)$$\begin{array}{cc} {\dot{V}}_{3}\leq & -W_{3}\left(\xi ,\eta ,e_{2},e_{3}\right)-e_{2}z_{a}+e_{3}{\dot{z}}_{a}-e_{2}\epsilon +0.2758\epsilon_{r}+e_{3}[k_{3}e_{3}+e_{2}+k_{2}{\hat{\dot{e}}}_{2}+\dot{\eta }+p\left(\hat{\varpi }\right)\\ & [{\bar{A}}_{as}(u-\gamma A_{as}x_{2}x_{3}v_{as}^{-1}+d_{max}tanh\left(\frac{e_{3}{\bar{A}}_{as}p\left(\hat{\varpi }\right)d_{max}}{\epsilon_{r}}\right)+\dot{Q}\left(\gamma -1\right){\left(s_{p}v_{as}\right)}^{-1})-{\dot{\hat{F}}}_{\delta }\\ & -\alpha f{\hat{\dot{x}}}_{2}\left(1-tanh^{2}\left(\alpha x_{2}+\beta sign\left(x_{1}\right)\right)\right)-c{\hat{\dot{x}}}_{2}]+\ddot{\xi }+\dot{p}\left(\hat{\varpi }\right)\left(F_{as}-F_{d}-{\hat{F}}_{\delta }\right)-{\dddot{x}}_{d}+k_{1}\ddot{\eta }]. \end{array}$$

Now, we choose the delay-free control input as
(37)$$\begin{array}{cc} u= & {\bar{A}}_{as}^{-1}[p{\left(\hat{\varpi }\right)}^{-1}\left(-k_{3}e_{3}-e_{2}-\dot{p}\left(\hat{\varpi }\right)\left(F_{as}-F_{d}-{\hat{F}}_{\delta }\right)+{\dddot{x}}_{d}-k_{1}\ddot{\eta }-\ddot{\xi }-k_{2}{\hat{\dot{e}}}_{2}-\dot{\eta }\right)\\ & +\alpha f{\hat{\dot{x}}}_{2}\left(1-tanh^{2}\left(\alpha x_{2}+\beta sign\left(x_{1}\right)\right)\right)+c{\hat{\dot{x}}}_{2}+{\dot{\hat{F}}}_{\delta }]-d_{max}tanh\left(\frac{e_{3}{\bar{A}}_{as}d_{max}p\left(\hat{\varpi }\right)}{\epsilon_{r}}\right)\\ & +\gamma A_{as}x_{2}x_{3}v_{as}^{-1}-\dot{Q}\left(\gamma -1\right){\left(s_{p}v_{as}\right)}^{-1}. \end{array}$$

Substituting (Equation 37) into (Equation 36), we obtain
(38)$$\begin{array}{c} {\dot{V}}_{3}\leq -k_{1}\eta^{2}-\alpha \xi^{2}-k_{2}e_{2}^{2}-k_{3}e_{3}^{2}+0.2785\epsilon_{r}-e_{2}z_{a}+e_{3}{\dot{z}}_{a}. \end{array}$$
where ${\dot{z}}_{a}=u\left(t\right)-u\left(t-\tau_{k}\right)$.

**Remark 3.** 
*The introduced projection function (Equation 23) is of class $C^{1}$, and ensures that $p(\hat{\varpi })$ satisfies $p\left(\hat{\varpi }\right)\in \left[\varpi_{min}-\varpi ,\;\varpi_{max}-\varpi \right]$. Setting $m_{min}=1/\left(\varpi_{max}+\varpi \right)$ and $m_{max}=1/\left(\varpi_{min}-\varpi \right)$, we can conclude that the designed delayed-free input (Equation 37) is always well defined since $p(\hat{\varpi })$ is always nonzero.*


In summary, the main result of the height tracking problem is summarized in the following theorem.

**Theorem 1.** 
*Let $x_{d}\in R^{3}$ in (Equation 4) be the proposed height whose time derivatives are bounded and continuous. By considering the closed-loop system obtained by the following control law (Equation 37), the RBFNN approximator online compensates for MRD force in (Equation 11), and the estimated sprung mass in (Equation 28). Then, the tracking errors $e_{i}=\left[e_{1},e_{2},e_{3}\right]$, given by (Equation 4), (Equation 19), and (Equation 31) can eventually converge to a small neighborhood of zero in the presence of modeling error $d_{n}$ and input delay τ, achieving uniform ultimate boundedness.*


We start the proof by using mean-value theorem to (Equation 38), leading to
(39)$${\dot{V}}_{3}\leq -k_{1}\eta^{2}-\alpha \xi^{2}-\theta_{2}e_{2}^{2}-\theta_{3}e_{3}^{2}-\frac{1}{4}\left(z_{a}^{2}-{\dot{z}}_{a}^{2}\right)+0.2785\epsilon_{r},$$
where $\theta_{2}$, $\theta_{3}$ are positive constants. Furthermore, we define a differentiable positive-definite functional as
(40)$$V=V_{3b}+\kappa V_{LK},$$
where λ is a positive constant, $V_{LK}$ is a Lyapunov–Krasovskii function, given as
(41)$$\begin{array}{c} V_{LK}=\int_{t-\tau_{k}}^{t}\left(\int_{s}^{t}{\left|u\left(\theta \right)\right|}^{2}d\theta )\right)ds, \end{array}$$
whose derivation derivative yields
(42)$$\begin{array}{cc} \dot{V}= & -k_{1}\eta^{2}-\alpha \xi^{2}-\theta_{2}e_{2}^{2}-\theta_{3}e_{3}^{2}-\frac{1}{4}\left(z_{a}^{2}-{\dot{z}}_{a}^{2}\right)+0.2785\epsilon_{r}+\kappa \tau_{k}{\left|u\right|}^{2}-\kappa \int_{t-\tau_{k}}^{t}{\left|u\left(\theta \right)\right|}^{2}d\theta . \end{array}$$

If the sufficient conditions in (Equation 41) are satisfied, then the following inequality can be defined
(43)$$\begin{array}{c} \left|u\right|\leq \phi +k_{3}\left|e_{3}\right|, \end{array}$$
where ϕ is a known positive constant, and using the fundamental inequality it gives us
(44)$$\begin{array}{cc} \kappa \tau_{k}{\left|u\right|}^{2} & =\kappa \tau_{k}\phi^{2}+2k_{3}\kappa \tau_{k}e_{3}\phi +\kappa \tau_{k}k_{3}^{2}e_{3}^{2}\\ \; & \leq \left(k_{3}+1\right)\kappa \tau_{k}\phi^{2}+\kappa \tau_{k}k_{3}^{2}e_{3}^{2}. \end{array}$$

Utilizing the Cauchy–Schwarz inequality, the integral in (Equation 41) can be upper bounded as
(45)$$\begin{array}{cc} -\kappa \int_{t-\tau_{k}}^{t}{\left|u\left(\theta \right)\right|}^{2}d\theta & \leq -\frac{\kappa }{2\tau_{k}}|z_{a}{|}^{2}-\frac{\kappa }{2}\int_{t-\tau_{k}}^{t}{\left|u\left(\theta \right)\right|}^{2}d\theta \\ -\frac{\kappa }{2}\int_{t-\tau_{k}}^{t}{\left|u\left(\theta \right)\right|}^{2}d\theta & \leq -\frac{\kappa }{2\tau_{k}}\int_{t-\tau_{k}}^{t}\left(\int_{s}^{t}{\left|u\left(\theta \right)\right|}^{2}d\theta \right)ds. \end{array}$$

Substituting (Equation 44), and (Equation 45) into (Equation 41), $\dot{V}$ is upper bounded as
(46)$$\begin{array}{c} \dot{V}=-k_{1}\eta^{2}-\alpha \xi^{2}-\Phi_{2}e_{2}^{2}-\Phi_{3}e_{3}^{2}-\Phi_{4}z_{a}^{2}-\Phi_{LK}V_{LK}+\Phi_{e}, \end{array}$$
where $\Phi_{2}=k_{2}+\frac{1}{4}$, $\Phi_{3}=k_{3}-\frac{1}{4}-\kappa \tau_{k}k_{3}^{2}$, $\Phi_{4}=\frac{1}{4}+\frac{\kappa }{2\tau_{k}}$, $\Phi_{LK}=\frac{\kappa }{2\tau_{k}}$, $\Phi_{e}=0.2785\epsilon_{r}+$$\left(k_{3}+1\right)\kappa \tau_{k}\phi^{2}$, and then setting $z={\left[\eta ,\xi ,e_{2},e_{3}\right]}^{T}$, ${\dot{V}}_{3}$ can be further upper bounded by
(47)$$\dot{V}\leq -k_{min}{∥z∥}^{2}+\Phi_{e},$$
where the auxiliary constant $k_{min}\in R^{3}$ is defined as
$$k_{min}≜min\left\{k_{1},\alpha ,\Phi_{2},\Phi_{3},\Phi_{4}\right\}.$$

Consequently, the above equation is ultimately negative for
(48)$$z>Z=\sqrt{\frac{\Phi_{e}}{k_{min}}}.$$

**Remark 4.** 
*According to Equation (Equation 48), the ultimate boundary size can be made very small by choosing a larger value of gain $k_{3}$. For arbitrary intervals of delay or arbitrarily large uncertain masses, the control gain required to satisfy the sufficient conditions in (Equation 28) and (Equation 44) depends in its size on the external level of road excitation and the inherent error ϕ resulting from u. Thus, to obtain a smaller error, a smaller α and a larger arbitrary control gain $k_{1},k_{2},k_{3}$ need to be used. However, it is necessary to find the appropriate balance between tracking accuracy and oscillator strength to avoid the controller causing unwanted oscillations, divergences, etc.*


**Remark 5.** 
*This work stabilises the conservativeness and computational complexity of the system by constructing a Lyapunov function that satisfies uncertain quality information and input delay information (as in (Equation 26) and (Equation 41)), and uses some suitable boundary techniques that can conservatively reduce the desired result to a minor value.*


## 5. Simulation Validation

To verify the effectiveness of proposed control strategy, in this section, by using the toolbox provided by AMESim(2021.2), a virtual plant of quarter vehicle with the AAS system is established. To obtain more realistic simulation results, we use a built-in secondary development component, Submodel Editor, to create and update the mathematical logic embedded in the MRDs and the mass blocks. In addition, the mathematical model and the proposed controller for the MRD-AAS system are programmed in Matlab/Simulink(2019(b)). Figure 5 shows the control block diagram for co-simulation. Major parameters used in the co-simulation are given in Table 2. Following [5], the control law $u_{\tau }$ is converted into the required changes of air mass ${\dot{m}}_{des}$, given by
(49)$${\dot{m}}_{des}=\left\{\begin{array}{cc} \frac{s_{p}v_{as}u_{\tau }}{\gamma RT_{s}}, & \mathrm{if}\;u_{\tau }>0\\ -\frac{s_{p}v_{as}u_{\tau }}{\gamma RT_{s}}, & \mathrm{otherwise}. \end{array}\right.$$

**Remark 6.** 
*The secondary development programme in AMESim, mentioned in [23,24] of this work to satisfy specific simulation demands, creates the required application libraries and software (as in (Equation 3) and (Equation 21)) flexibly through secondary development using some suitable mathematical logic and modifying the output and input port signals for specific demands and computational complexity.*


The corresponding PWM duty cycle $\left(D_{c}\right)$ of the chosen virtual control input *u* can be viewed as the control signal which can be determined as
(50)$$\left\{\begin{array}{c} \mathrm{Pro}.I:D_{c}=min\left(\frac{s_{p}v_{as}u_{\tau }}{{\dot{m}}_{in}\gamma RT_{s}},1\right)\\ \mathrm{Pro}.\mathrm{II}:D_{c}=min\left(-\frac{s_{p}v_{as}u_{\tau }}{{\dot{m}}_{in}\gamma RT_{s}},1\right)\\ \mathrm{Pro}.\mathrm{III}:D_{c}=0. \end{array}\right.$$

In order to demonstrate the height tracking performance of the proposed control system in the presence of sprung mass variations and time-varying input delay simultaneously, the uncertain sprung mass is divided into five stages: (i) 0 to $t_{1}$: m=300(kg); (ii) $t_{1}$ to $t_{2}$: m=330(kg); (iii) $t_{2}$ to $t_{3}$: m=360(kg); (iv) $t_{3}$ to $t_{4}$: m=330(kg); and (v) $t_{4}$ to end: m=300(kg). Meanwhile, with a current of I=0(A), the input time delay is divided into three groups: $\tau_{1}=\tau_{k_{1}}+5sin\left(\pi t\right)\;(\mathrm{ms})$, $\tau_{2}=\tau_{k_{2}}+5cos\left(0.5\pi t\right)\;(\mathrm{ms})$, $\tau_{3}=\tau_{k_{3}}+5cos\left(\pi t\right)*sin\left(0.5\pi t\right)\;(\mathrm{ms})$, $\tau_{4}=\tau_{k_{2}}+5cos\left(0.5\pi t\right)\;(\mathrm{ms})$, $\tau_{5}=\tau_{k_{1}}+5sin\left(\pi t\right)\;(\mathrm{ms})$, with $\tau_{k_{1}}=10\;\;(\mathrm{ms})$, $\tau_{k_{2}}=20\;\;(\mathrm{ms})$, $\tau_{k_{3}}=30\;\left(\mathrm{ms}\right)$, and the disturbance occurring on the system is set as $d_{n}=0.018cos\left(0.01t\right)$. The co-simulation is carried out on a rough road corresponding to class B of an ISO road profile with a driving speed of 40 (km/h).

The time evolution of height $z_{s}$ and tracking error $e_{1}$ with uncertain mass $m_{s}$ and time-varying delays of $\tau_{1}$, $\tau_{2}$ and $\tau_{3}$ under the driving speeds of 40(km/h) and 100(km/h) are displayed in Figure 6a,b, respectively. The proposed delay-free control law enables the height to converge towards the desired value within 2(s). The estimated error of sprung mass $\tilde{m}$ is bounded by the range of 0.5(kg), as shown in Figure 7. Correspondingly, the tracking error $e_{1}$ undergoes new transients at $t=t_{i},\;i=1,2,3,4$, but always converges to a neighborhood of zero as time increases and is ultimately bounded by a tolerant value 1(mm). Moreover, the designed RBFNN approximator could compensate for the non-simulated dynamics of the MRD within the lagged time range 30(ms), as shown in Figure 8.

In order to further demonstrate the advantages of the proposed control strategy without consideration of uncertain mass and time-varying input delay, Figure 9 shows that the proposed controller and the method presented in [25] could drive the height tracking error close to zero under the same co-simulation parameters and road disturbances. The comparison results obtained from both two controllers are given in Table 3. It is noted that the root mean square (RMS) and response time (RT) of the ride height and the adjustment time are decreased by using the proposed controller. Simulation results in Figure 9 and the performance comparison in Table 3 indicate that the proposed control technique outperforms the HMPC given in [25].

## 6. Conclusions

This paper provided an innovative neural network-driven approach to tackle the issue of adjusting vehicle height with an MRD-AAS system in the presence of unmodeled dynamics of MRD, uncertain mass, and time-varying input delays. A RBFNN approximator was designed with the adaptive gradient descent learning algorithm to compensate for the concentrated modeling error of the MRD caused by the unmodeled dynamics. A delay-free control strategy was synthesized to enable the ride height converge to an arbitrarily small neighborhood of the preset desired height. The designed RBFNN approximator and mass law were integrated into the delay-free control input, achieving uniform ultimate boundedness. Co-simulation results were given to validate the performance and effectiveness of the proposed control strategy.

With respect to future work, a deep reinforcement learning-based optimal MRD-AAS control system will be designed to maintain leveling and posture motion control while improving performance in terms of isolating vibrations holding capacity and handling stability.

## Figures and Tables

**Figure 1 sensors-24-00156-f001:**
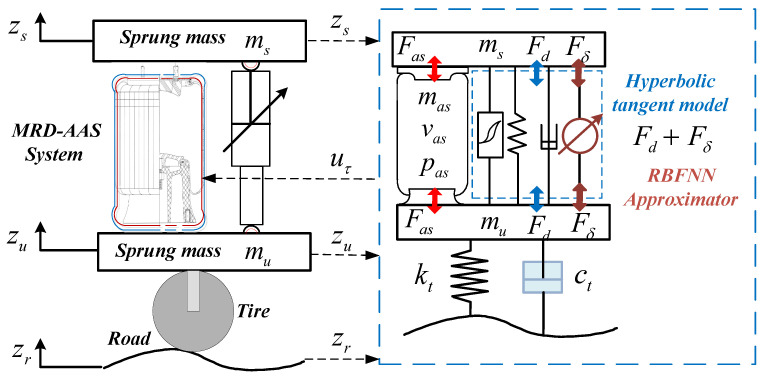
Schematic of quarter vehicle with MRD-AAS.

**Figure 2 sensors-24-00156-f002:**
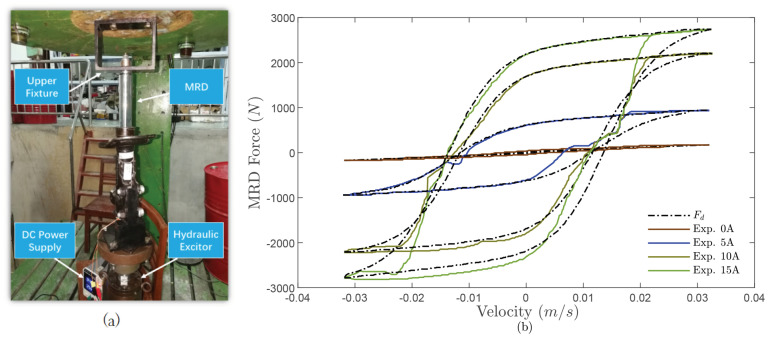
MRD external dynamic tests: (**a**) Test bench; (**b**) Measured dynamic characteristics.

**Figure 3 sensors-24-00156-f003:**
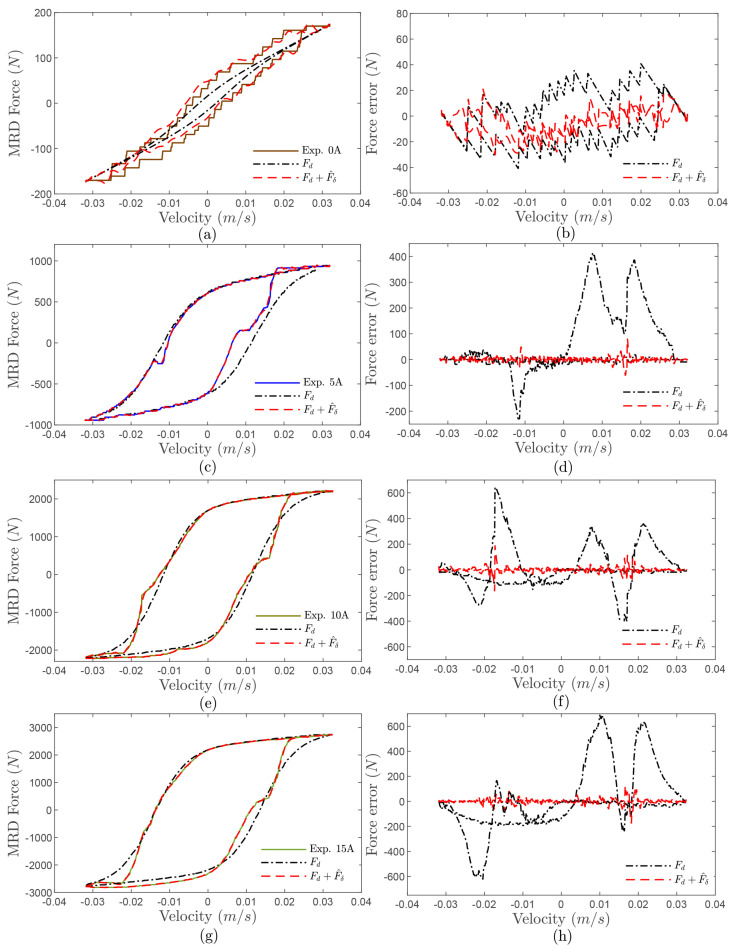
RBFNN approximator performance: (**a**) MRD force at 0 A; (**b**) RBFNN approximated error at 0 A; (**c**) MRD force at 1 A; (**d**) RBFNN approximated error at 1 A; (**e**) MRD force at 2 A; (**f**) RBFNN approximated error at 2 A; (**g**) MRD force at 3 A; (**h**) RBFNN approximated error at 3 A.

**Figure 4 sensors-24-00156-f004:**
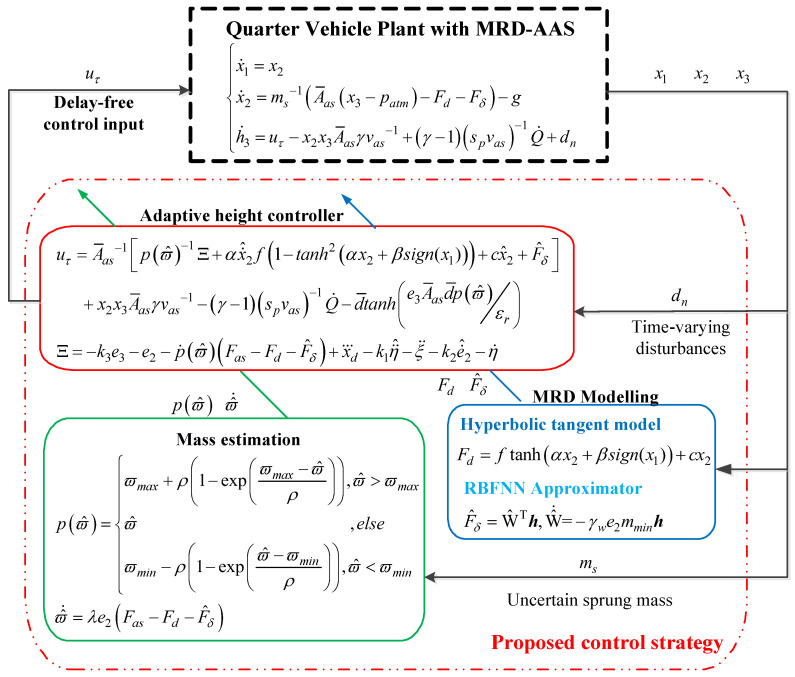
Block diagram of proposed control strategy.

**Figure 5 sensors-24-00156-f005:**
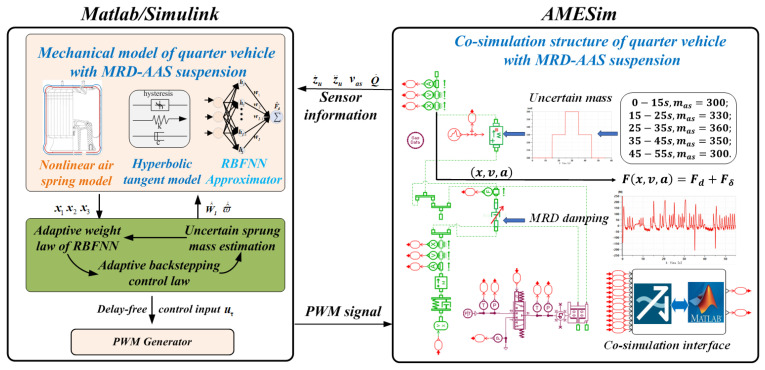
Control block diagram of co-simulation.

**Figure 6 sensors-24-00156-f006:**
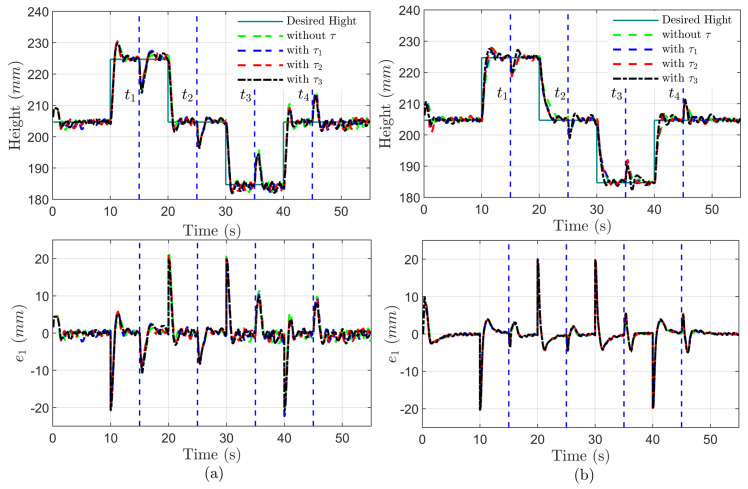
Height tracking performance of quarter vehicle MRD-AAS in co-simulation: (**a**) At speed of 40 km/h; (**b**) At speed of 100 km/h.

**Figure 7 sensors-24-00156-f007:**
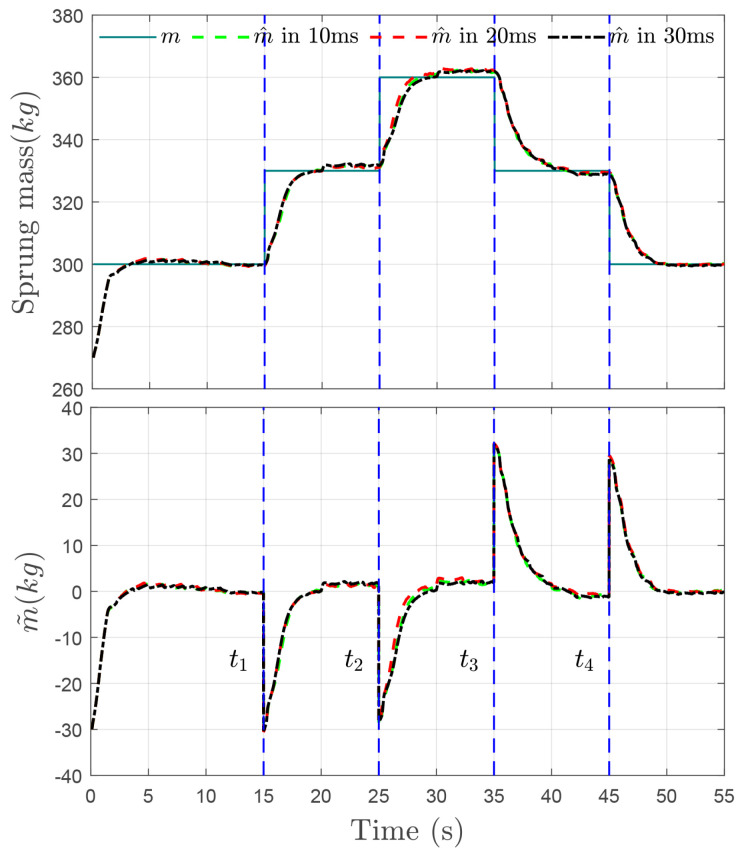
Time evolution of the estimated mass $\hat{m}$ and mass estimation error $\tilde{m}$ with input delays $\tau_{1}$, $\tau_{2}$, $\tau_{3}$.

**Figure 8 sensors-24-00156-f008:**
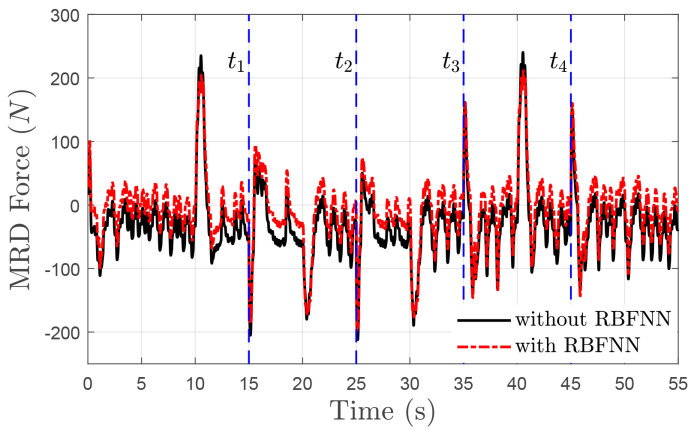
MRD force comparison with time-varying input delay $\tau_{3}$ in co-simulation.

**Figure 9 sensors-24-00156-f009:**
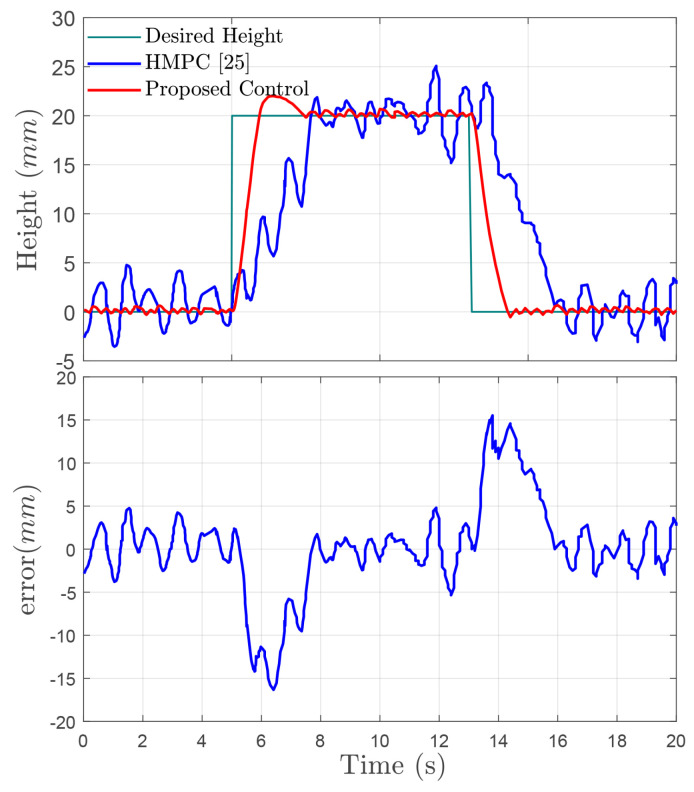
Performance comparison and height error on a quarter vehicle in co-simulation. HMPC in [25].

**Table 1 sensors-24-00156-t001:** Nomenclature.

Symbol	Description	Symbol	Description
$z_{s}\;(m)$	sprung mass displacement	$D_{c}$	duty cycle of PWM signal
$z_{u}\;(m)$	unsprung mass displacement	$T_{p}\;(t)$	PWM signal period
$z_{r}\;(m)$	road disturbance	${\dot{m}}_{in}\;(\mathrm{kg}\;\cdot \;s^{-1})$	inflating air mass rate
$m_{s}\;(\mathrm{kg})$	sprung mass	${\dot{m}}_{out}\;(\mathrm{kg}\;\cdot \;s^{-1})$	deflating air mass rate
$m_{u}\;(\mathrm{kg})$	unsprung mass	${\dot{m}}_{des}\;(\mathrm{kg}\;\cdot \;s^{-1})$	desired air mass change
$A_{as}\;(m^{2}$)	surface area of air spring	f(MRD)	hysteresis loop scaling factor
${\bar{A}}_{as}\;(m^{2})$	effective area of air spring	α(MRD)	hysteresis loop scaling factor
$v_{as}\;(m^{3})$	air spring volume	β(MRD)	hysteresis loop half-width factor
$F_{d}\;(N)$	MRD force of hyperbolic model	c(MRD)	MRD damping coefficient
γ	adiabatic index of ideal gases	$\tau_{k}\;(\mathrm{ms})$	delay-time
*R*	ideal-gases constant	$x_{d}\;(m)$	desired height
$\dot{Q}\;(J\;\cdot \;s^{-1})$	heat transfer rate	*u*	actuator command
$P_{u}\;(\mathrm{Pa})$	upstream pressure	$k_{1},k_{2},k_{3}$	positive constants
$P_{dn}\;(\mathrm{Pa})$	downstream pressure	${\tilde{m}}_{s}\;(\mathrm{kg})$	mass estimation error
$p_{atm}\;(\mathrm{MPa})$	atmosphere pressure	${\hat{m}}_{s}\;(\mathrm{kg})$	mass estimation
$P_{cr}\;(\mathrm{Pa})$	critical pressure ratio	$\gamma_{w}$	constant coefficient
$c_{q}$	deflating air mass rate	λ	constant coefficient
T(K)	air temperature	$F_{\delta }\;(N)$	modeling error of MRD
κ	constant coefficient	${\tilde{F}}_{\delta }\;(N)$	estimation error of MRD force
ϱ	constant coefficient	${\hat{F}}_{\delta }\;(N)$	estimation of MRD force

**Table 2 sensors-24-00156-t002:** Parameter used in simulation.

Parameter	Value	Parameter	Value
$A_{as}$	$0.0072\;\left(m^{2}\right)$	$\alpha_{1}$	1.8506
$A_{heat}$	$500\;\left(J/Ksm^{2}\right)$	$\alpha_{2}$	100.48
*g*	$9.807\;\left(m/s^{2}\right)$	$\beta_{1}$	0.0819
γ	1.4	$\beta_{2}$	0.5245
$P_{atm}$	1.0133(bar)	$c_{1}$	651.23
$s_{p}$	$10^{5}$	$c_{2}$	438.3
$z_{0}$	0.2047(m)	$k_{1}$	1
$p_{0}$	5.4(bar)	$k_{2}$	5
$S_{x}$	$10^{-5}\;\left(m^{2}\right)$	$k_{3}$	10
$C_{q}$	$8\times 10^{-2}$	$\epsilon_{1}$	$1\times 10^{-2}$
*R*	287.1	$z_{0i}$	0.05(m)
*v*	40(km/h)	*I*	0 A
$G_{0}$	$0.64\times 10^{-4}$	$d_{max}$	0.02
$m_{u}$	30(kg)	tolerance	$1\times 10^{-5}$
$f_{1}$	160.88	α	1
$f_{2}$	32.8981	*℘*	190

**Table 3 sensors-24-00156-t003:** Comparison of controller tracking performance.

	RMSE of $z_{1}$			RT of $z_{1}$		
**Times Range (s)**	**Method in [25]**	**Proposed Method**	**Improvement ^1^**	**Method in [25]**	**Proposed Method**	**Improvement ^1^**
0 to 5 s	0.6273	0.1521	75.75%	0.2549	0.1459	42.76%
5 to 10 s	7.3058	2.7969	61.72%	7.6781	5.7588	25.00%
10 to 13 s	0.2974	0.1858	37.53%	/	/	/
13 to 15 s	17.0576	6.8637	59.76%	16.1348	14.3173	11.26%
15 to 20 s	1.0783	0.1626	84.92%	/	/	/

Improvement ^1^ = (Proposed method—Method in [25])/Method in [25].

## Data Availability

The datasets generated during and/or analyzed during the current study are not publicly available but are available from the corresponding author upon reasonable request.

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
