# Peer review of "Neural Network-Based Adaptive Height Tracking Control of Active Air Suspension System with Magnetorheological Fluid Damper Subject to Uncertain Mass and Input Delay"

_sensors, 2023, doi:10.3390/s24010156_

Round 1

Reviewer 1 Report

Comments and Suggestions for Authors

In this paper, a novel robust adaptive neural network-based control framework is developed for the MRF-AAS system, which consists of a radial basis function neural network approximator, a projector-based estimator and a time-delay compensator. It solves the problem address the height tracking control problem for nonlinear MRF-AAS systems with dynamic characteristics, uncertain parameter variation and time-varying input delay effectively.

The manuscript is very organized, well-written, and well-presented. However, some revisions are needed before it is accepted for publication in the journal. The followings are the comments or issues existing in this version.

1. There are many grammar errors in the paper. Such as, in page 2 line 63, “In order to deal with the with the highly-nonlinear”; in page 4 line 125 “As display in figure 1”; and so on.

2. What’s meaning of the Fδ in figure 1?

3. What does the “Experiments force” in figure 2 mean and how do you get it?

4. In figure 2, a current of 15A is applied on the MR damper, how about of the power consumption of the MR damper? And the properties of the MR fluid will change at this conditions.

Author Response

1. There are many grammar errors in the paper. Such as, in page 2 line 63, “In order to deal with the with the highly-nonlinear”; in page 4 line 125 “As display in figure 1”; and so on.

Authors: We appreciate the reviewer for pointing out these mistakes, we have corrected typos and grammar mistakes.

2. What’s meaning of the Fδin figure 1?

Authors: In response to the reviewer’s comment, $F_{\delta}$ represents the concentrated modeling error of the MRD in Page 6 Line 159.

3. What does the “Experiments force” in figure 2 mean and how do you get it?

Authors: In response to the reviewer’s comment, the “Experiments force” is the MRD force, which is obtained through experimental tests. The follow sentences are added in the revised manuscript.

To verify the effectiveness of the constructed MRD model, a MRD test bench is fabricated as Figure 2(A) displayed, which consists of upper fixture, DC power supply, hydraulic exciter and MRD. As Figure 2(B) shown the measured external characteristics of the employed MRD at the current from 0 to 3(A) $ are collected and given. Moreover, Figures 2(a~h) show the comparison of a hyperbolic tangent model with and without the designed RBFNN approximator. It can be found that the designed RBFNN approximator could compensate for the concentrated modeling errors and keep the approximated error in a bound of 20(N)}. Therefor e, the constructed MRD model combined the hyperbolic tangent model with the designed RBFNN approximator is then applied in the following control system design.

4. In figure 2, a current of 15A is applied on the MR damper, how about of the power consumption of the MR damper? And the properties of the MR fluid will change at these conditions.

Authors: Thanks for the reviewer’s comment, some mistakes are made about the current in Figure 2. The current of 0~3A, not 0~15A, is applied on the MR damper. The power consumption of the MRD is given as

where  denotes the power consumption,  is applied current, and  is the resistance of the coil wire. Practically, the magnetorheological fluid dampers could offer an outstanding capability in semiactive vibration control due to excellent dynamical features such as fast response, environmentally robust characteristics, large force capacity, low power consumption, and simple interfaces between electronic input and mechanical output [A1].

[A1] Zhu X, Jing X, Cheng L. Magnetorheological fluid dampers: A review on structure design and analysis [J]. SAGE Publications, 2012(8). DOI:10.1177/1045389x12436735.

Additionally, as Figure A1 shown, in an MR damper the piston includes coils capable of delivering a magnetic field in the orifices. In these terms, the piston may be viewed as a “magnetorheological valve” and the damping is the result of the friction between the fluid and the orifices. When subject to a magnetic field caused by the current , MR fluids change their viscosity

Reviewer 2 Report

Comments and Suggestions for Authors

This paper provided a novel neural network-based solution to address the problem of vehicle height tracking control with MRF-AAS system in the presence of unmodeled dynamics of MRD, uncertain mass, and time-varying input dealy. By employing the backstepping technique, a delay-free control strategy was proposed to drive the ride height to an arbitrarily small neighborhood of the preset desired height. A RBFNN approximator is developed with the adaptive gradient descent learning algorithm to compensate for the unmodeled dynamics of MRD. A projector-based updating estimation law is designed to estimate the uncertain sprung mass.

The problem considered by the authors is very relevant for today. The proposed algorithms on the basis of neural networks are also in the fairway of the development of science and technology. I believe that the concept proposed by the authors to solve the problem of tracking the height of the vehicle is quite original and deserves publication in the journal.

Author Response

Thanks for the comments of Reveiwer 2.

Reviewer 3 Report

Comments and Suggestions for Authors

The authors considered an original technical solution for a controlled car suspension based on active air systems with magnetorheological fluid damper. By employing the backstepping technique, a delay-free control strategy was proposed to drive the ride height to an arbitrarily small neighborhood of the preset desired height. A RBFNN approximator is developed with the adaptive gradient descent learning algorithm to compensate for the unmodeled dynamics of MRD. The proposed RBFNN approximator and mass estimation law are embedded into the delay-free control input, obtaining robust adaptive performance. Co-simulation results were presented to validate the performance and effectiveness of the proposed control strategy.

Several comments can be made on the article:

In the title, abstract, introduction and in the text of the article, the authors use the phrase “magnetorheological fluid active air suspension”. It seems that the suspension is based on a chemical suspension of magnetorheological fluid and air. It was correct to use the term “active air suspension system with magnetorheological fluid damper”.

In the introduction, the authors make a very ambitious statement: “The magnetorheological fluid active air suspension (MRF-AAS) provides a promising solution to fulfill the stringent requirements of chassis functions.” However, there are no references to experimental work proving the effectiveness of this combined solution compared to traditional controlled suspensions.

Some curves in the figure are designated as experimental for various values of current in the coil. The text of the article does not mention a single experiment conducted by the authors. If this is a numerical experiment, then you need to rename the curves on the graph and give its parameters in the description. If this is data from another work, then you need to insert a link to it.

The abscissa axes are not indicated in graph 2.

It is not entirely correct to present the MRD Force dependences for different current strengths in the electromagnet. It is necessary either to plot the dependence of the magnetic field strength on the current in the magnetizing coil, or to replace the current values with the magnetic field strength.

How was the demagnetizing field arising in a limited volume of magnetic fluid taken into account?

I did not see the parameters of the magnetic system and magnetorheological system used in the simulation and listed in Table 2. I recommend adding designations for these parameters to this table and making a link to the formula in which they are used.

After these minor changes, the article can be published.

Author Response

1. In the title, abstract, introduction and in the text of the article, the authors use the phrase “magnetorheological fluid active air suspension”. It seems that the suspension is based on a chemical suspension of magnetorheological fluid and air. It was correct to use the term “active air suspension system with magnetorheological fluid damper”.

Response:Thanks for the reviewer’s comment, the phrase “magnetorheological fluid active air suspension” are replaced with “active air suspension system with magnetorheological fluid damper” throughout the paper.

2. In the introduction, the authors make a very ambitious statement: “The magnetorheological fluid active air suspension (MRF-AAS) provides a promising solution to fulfill the stringent requirements of chassis functions.” However, there are no references to experimental work proving the effectiveness of this combined solution compared to traditional controlled suspensions.

Thanks for the reviewer’s comment, this sentence is rewritten as: “The active air suspension system with magnetorheological fluid damper (MRD-AAS) provides an optional solution to meet the stringent requirements of chassis functions” in the revised manuscript.

To verify the effectiveness of the constructed MRD model, a MRD test bench is fabricated as Figure 2(A) displayed, which consists of upper fixture, DC power supply, hydraulic exciter and MRD. As Figure 2(B) shown the measured external characteristics of the employed MRD at the current from 0 to 3(A) $ are collected and given. Moreover, Figures 2(a~h) show the comparison of a hyperbolic tangent model with and without the designed RBFNN approximator. It can be found that the designed RBFNN approximator could compensate for the concentrated modeling errors and keep the approximated error in a bound of 20(N)}. Therefor e, the constructed MRD model combined the hyperbolic tangent model with the designed RBFNN approximator is then applied in the following control system design.

3. Some curves in the figure are designated as experimental for various values of current in the coil. The text of the article does not mention a single experiment conducted by the authors. If this is a numerical experiment, then you need to rename the curves on the graph and give its parameters in the description. If this is data from another work, then you need to insert a link to it.

In response to the reviewer’s valuable comment, the MRD force is obtained by a series of experimental tests on the test bench as Figure A1 displayed. In order to describe the dynamic characteristics of MRD, experimental tests have been done on the test bench with the current A from 0 to 3A. The experimental results are collected and shown in Figure 2.

4. The abscissa axes are not indicated in graph 2.

Thanks for the reviewer’s comment, the abscissa axes are added, and Figure 2 is modified in the revised manuscript.

5. It is not entirely correct to present the MRD Force dependences for different current strengths in the electromagnet. It is necessary either to plot the dependence of the magnetic field strength on the current in the magnetizing coil, or to replace the current values with the magnetic field strength.

In response to reviewer’s comment, this paper focus on practically addressing the ride height tracking control with MRD-AAS system by considering the MRD output force, which is determined by the amount of input current. Therefore, experimental tests have been done on the test bench with the current from 0 to 3A to mapping the relationship of the dynamic characteristics of MRD and input current.

6. How was the demagnetizing field arising in a limited volume of magnetic fluid taken into account?

In response to reviewer’s comment, as Figure A1 shown, In an MR damper the piston includes coils capable of delivering a magnetic field in the orifices. B is the magnetic field, which is determined by the current . The MRD output force is the result of the friction between the fluid and the orifices.

7. I did not see the parameters of the magnetic system and magnetorheological system used in the simulation and listed in Table 2. I recommend adding designations for these parameters to this table and making a link to the formula in which they are used.

In response to reviewer’s comment, a hyperbolic tangent model is employed and combined with a designed RBFNN approximator to effectively describe the nonlinear hysteretic characteristics of the MRD current-force, expressed by Equations (3), (4) and (11). All the designations for these parameters are listed in Table 2.
